# Two oxytocin analogs, N-(p-fluorobenzyl) glycine and N-(3-hydroxypropyl) glycine, induce uterine contractions ex vivo in ways that differ from that of oxytocin

Stanislav M. Cherepanov[1]*, Teruko Yuhi[1], Takashi Iizuka[2], Takashi Hosono[2], Masanori Ono[2], Hiroshi Fujiwara[2], Shigeru Yokoyama[2], Satoshi Shuto[3], Haruhiro Higashida[1]

**1** Department of Basic Research on Social Recognition and Memory, Research Center for Child Mental Development, Kanazawa University, Kanazawa, Ishikawa, Japan, **2** Department of Obstetrics and Gynecology, Kanazawa University Graduate School of Medical Science, Kanazawa University, Kanazawa, Ishikawa, Japan, **3** Faculty of Pharmaceutical Sciences and Center for Research and Education on Drug Discovery, Hokkaido University, Sapporo, Hokkaido, Japan

* stascherneuro808@gmail.com

## Abstract

Contraction of the uterus is critical for parturient processes. Insufficient uterine tone, resulting in atony, can potentiate postpartum hemorrhage; thus, it is a major risk factor and is the main cause of maternity-related deaths worldwide. Oxytocin (OT) is recommended for use in combination with other uterotonics for cases of refractory uterine atony. However, as the effect of OT dose on uterine contraction and control of blood loss during cesarean delivery for labor arrest are highly associated with side effects, small amounts of uterotonics may be used to elicit rapid and superior uterine contraction. We have previously synthesized OT analogs 2 and 5, prolines at the $7^{th}$ positions of which were replaced with N-(p-fluorobenzyl) glycine [thus, compound 2 is now called fluorobenzyl (FBOT)] or N-(3-hydroxypropyl) glycine [compound 5 is now called hydroxypropyl (HPOT)], which exhibited highly potent binding affinities for human OT receptors *in vitro*. In this study, we measured the *ex vivo* effects of FBOT and HPOT on contractions of uteri isolated from human cesarean delivery samples and virgin female mice. We evaluated the potency and efficacy of the analogs on uterine contraction, additivity with OT, and the ability to overcome the effects of atosiban, an OT antagonist. In human samples, the potency rank judged by the calculated $EC_{50}$ (pM) was as follows: HPOT (189) > FBOT (556) > OT (5,340) > carbetocin (12,090). The calculated Emax was 86% for FBOT and 75% for HPOT (100%). Recovery from atosiban inhibition after HPOT treatment was as potent as that after OT treatment. HPOT showed additivity with OT. FBOT (56 pM) was found to be the strongest agonist in virgin mouse uterus. HPOT and FBOT demonstrated high potency and partial agonist efficacy in the human uterus. These results suggested that HPOT and FBOT are highly uterotonic for the human uterus and performed better than OT, indicating that they may prevent postpartum hemorrhage.

**Data Availability Statement:** All relevant data are within the paper and its Supporting Information files.

**Funding:** The authors received no specific funding for this work.

**Competing interests:** The authors have declared that no competing interests exist.

## Introduction

Oxytocin (OT) is a nonapeptide hormone that is mainly synthesized in the neurons of the supraoptic and paraventricular nuclei of the hypothalamus [1]. OT, released into circulation by the posterior pituitary gland, induces uterine contractions and milk ejection during delivery and lactation [2]. Additionally, OT in the central nervous system enhances social communication, such as social behavior, recognition, and memory, and functions as a neuromodulator [3].

We have previously reported three OT analogs, named lipidated-OTs (LOTs), in which a palmitoyl group was linked to the cysteine terminal amino group and/or the tyrosine aromatic hydroxyl group of OT [4]. These OT analogs displayed curative and long-lasting effects on social impairments in CD38 and CD157 knockout mice after a single intraperitoneal injection at a dose of 0.3–3 ng/g body weight [4–6]. In addition, we synthesized new highly potent OT analogs **2** and **5**; in particular, compound **2** possesses agonistic ability superior to that of the natural agonist, OT. In other words, compound 2 is a superagonist, which is the first example for a human OT receptor [7].

Historically, fluorination of ligands has improved their pharmacological and pharmacokinetic properties [8,9]. The unique transformation of the proline residue at position seven of compound 2 to an N-(p-fluorobenzyl) glycine residue yielded FBOT, while that in compound 5 to N-(3-hydroxypropyl) glycine yielded HPOT [7]. However, whether these behave as analogs on OT receptors expressed in tissues, such as the uterus, remains unknown.

Contraction of the uterus is critical in late parturient processes, such as fetal and placental expulsion [10]. Insufficient uterine tone resulting in atony can potentiate postpartum hemorrhage, and it is known to cause adverse outcomes for parturients [11]. Moreover, uterine atony is a major risk factor and the main cause of maternal deaths worldwide [12–14]. The prevention and management of postpartum hemorrhage during vaginal or cesarean delivery is critical, particularly in patients with anemia or preeclampsia who cannot tolerate even minimal blood loss [15].

Many studies have examined the dosage and outcome of using intravenous uterotonics, including OT [16–21]. Currently, OT is recommended for use in combination with other uterotonics for treating refractory uterine atony, although the dosage remains uncertain [22]. However, OT receptor desensitization due to exogenous OT administration during cesarean section [15] leads to use of high OT doses (approximately nine times higher) for inducing effective uterine contraction and labor arrest [23]. Hence it is critical to attain a favorable balance between effective uterine contractions, reduced hemorrhage, and analogue-induced adverse events induced by high doses of OT. Therefore, the use of alternative uterotonic agents, instead of overdosage of synthetic OT, may result in superior uterine contraction and control blood loss during cesarean delivery for labor arrest.

Misoprostol, a prostaglandin derivative, is considered as effective as OT in controlling intraoperative and postoperative bleeding [24,25]. In addition, a combination of misoprostol and OT has been reported to be more effective than OT alone in reducing intraoperative and postoperative bleeding during caesarean section [26]. However, recent clinical trials have demonstrated limited effects of this combination as well as increase in risk of adverse effects such as fever, nausea, and vomiting [27].

Data obtained from laboring and non-laboring parturients have shown that carbetocin, a long-acting synthetic analogue of OT, is superior to OT in terms of requirement of additional uterotonics during delivery and prevention of postpartum hemorrhage [21,28]. Under elective caesaean delivery with carbetocin, more uterotonics were added only in rare cases [23]. This is possibly because of the extended mode of action of carbetocin, which is highly lipophilic and stable [29]. FBOT, HPOT, and carbetocin belong to the same family of OT analogs,

characterized by deamination of the terminal cysteine residue, with improved selectivity over vasopressin V1A and V1B receptors as shown in *in vitro* activity assays [7]. Thus, similar to carbetocin, intravenous administration of FBOT and HPOT may reduces the time required to achieve adequate uterine tone; subsequently, they can prevent atony after delivery of neonates, subsequently lowering the risk of postpartum hemorrhage. Furthermore, these analogs may have fewer side effects, such as hypertension, nausea, and vomiting.

In the present study, we aimed to evaluate the *ex vivo* effects of FBOT and HPOT on contraction of uteri isolated from human cesarean delivery samples. We determined the various contraction parameters induced by FBOT and HPOT and compared them with those induced by OT and carbetocin. Finally, we analyzed the agonistic effects of FBOT and HPOT in the uteri of virgin female mice.

## Methods and materials

### Tissue preparation from pregnant women

Myometrial tissues were collected from pregnant women undergoing scheduled elective cesarean section at term (36–38 weeks of pregnancy) prior to the onset of labor. All participating women were informed about the nature of the study in advance (**S1 Table**)**,** and informed written consent was obtained with the approval of the local research ethics committee (Ethics No 2018–130, 2896). Women with multiple pregnancies or medical conditions, such as diabetes, preeclampsia, or obstetric cholestasis, were not included in this study. There were 3 secondary and 20 primary cesarean sections out of 23. Myometrial biopsies were obtained from the upper margin of the incision made at the lower segment of the uterus. The sample size was approximately $5 \times 5 \times 10$ mm and they were stored in phosphate-buffered saline at 4˚C until dissection (**S1 Fig**). In the case of the secondary caesarean section, we performed a caesarean section with an incision cranial to the uterine incision scar of the previous caesarean section to minimize the presence of fibrous tissue. All samples were transferred to the laboratory for contractility experiments within 2–4 h of collection. The biopsies were dissected into eight longitudinal myometrial strips of 7 x 2 x 1 mm dimensions and mounted in a thermostatically controlled isolated chamber. Following delivery of the placenta, all women immediately received 5 units of OT (Syntocinon Novartis, Switzerland) via an intravenous line. OT administration is part of the standard care for the prevention of postpartum hemorrhage. Myometrial biopsies were excised within 3 min of OT administration.

### *Ex vivo* uterine contraction study

Stainless steel hooks were passed through the uterine segments in opposite directions on each end and mounted in the organ bath by fixing the bottom end, while the other end was attached to a lever connected to the recording apparatus, an isotonic transducer (TR210A, Isometric Teaching, ADInstruments Ltd., Australia). Muscle preparations were incubated in a 30 mL organ bath containing Tyrode's solution [8.0 g/L NaCl, 0.2 g/L KCl, 0.2 g/L CaCl$_2$, 0.1 g/L MgCl$_2$, 1.0 g/L NaHCO$_3$, 0.05 g/LNaH$_2$PO$_4$, and 1.0 g/L glucose (pH 7.4)] in which 95% O$_2$ and 5% CO$_2$ were constantly bubbled at a constant rate of 2–3 bubbles/s. The temperature of the solution was maintained at 37˚C. Thereafter, uterine movement was continuously recorded using the isotonic transducer coupled to a bridge pod and a bridge amplifier (PowerLab2/26 Data Acquisition System, ADInstruments) using a standard laboratory protocol [30]. The uterine preparations were allowed to equilibrate for at least 45 min by applying an initial resting force of 1 g. During this period, the uterine strips were washed with 30 mL fresh physiological solution every 15 min according to the method described by Oropeza et al. [31]. Between administration of different drugs and different doses of the same drug, the bathing

solution was flushed out from the organ bath and immediately replaced with a new solution containing the desired compound or the desired dose of the same compound (300 μL of solution contain a compound in concentration 100 times higher than the final concentration of the compound was added to 30 mL of the new solution). Subsequently, the responses of the uterine strips were recorded. The exposure time of each of the uterine strips to the drug was 15–20 min. The data were stored and subsequently retrieved using the LabChart Pro 6 (ADInstruments) software. Recording was performed at a sweep speed of 117 mm/s, deflection of 1000 mm, low path filter of 0.5 Hz, high path filter of direct current, and sensitivity of 20 μV. The area under the curve (AUC) of uterine contraction was recorded and calculated using LabChart Pro 6.

## Tissue preparation from mice

Female Slc:ICR mice (Institute of Cancer Research of the Charles River Laboratories, Inc., Wilmington, MA, USA) were obtained from Japan SLC, Inc. (Hamamatsu, Japan) through a local distributor (Sankyo Laboratory Service Corporation, Toyama, Japan). Mice were housed in a nursing cage in our laboratory under standard conditions (24°C; 12 h light/dark cycle, lights on at 08:00) with food and water provided ad libitum. All animal experiments were performed in accordance with the Fundamental Guidelines for the Proper Conduct of Animal Experiments and Related Activities in Academic Research Institutions under the jurisdiction of the Ministry of Education, Culture, Sports, Science, and Technology of Japan and were approved by the Committee on Animal Experimentation of Kanazawa University (ethics approval code AP-173824).

For studying uterine movement, adult virgin ICR female mice (8–10 week old) were anesthetized using intraperitoneal injection of 10% pentobarbital (30 mg/kg body weight). The abdomen was immediately opened and the uterine segments were removed via a transverse incision. After removal, the uterine horn of the mid portion was placed longitudinally in a 30 mL organ bath containing Tyrode's solution, as described above.

## Statistical analysis

Statistical significance was evaluated using GraphPad Prism 8 (GraphPad Software LLC, San Diego, CA, USA). The uterine contraction power due to the agonist activity of the OT analogs was fitted to three parameters ($EC_{50}$, Top, Bottom): dose response stimulation equation with variable slope and the following formula: $Y = Bottom + (Top-Bottom)/(1+10/(Log(EC_{50}-X))$, where Y is the percentage normalized contraction AUC. The bottom is the minimum percentage of response, top is the maximum percentage of response, X is the logarithm of the concentration of the compound, and the Hill slope of the Hill coefficient = 1 All compounds were tested three times in duplicates. Additivity was calculated according to the Bliss independence model [32] according to the formula CI = (EA + EB—EAxEB)/EAB. Where EA–response for oxytocin application, EB–response for tested compound application and EAB–response for mix of oxytocin and compound. Response is considered in the range $0 < E < 1$, where 0 is percentage of control contraction AUC and 1 is percentage of Emax contraction AUC. Antagonist response data were analyzed using one-way analysis of variance (ANOVA), and individual comparisons were performed using Bonferroni's multiple correction test. Statistical significance was set at $P < 0.05$.

## Results

### Effects of oxytocin analogs on contraction of human uterine muscles

Human uterine contractions were examined in samples isolated during cesarean delivery. Fig 1 shows representative traces of enhanced spontaneous contraction from the basal contraction condition induced by 1 nM each of OT, cabetocin, FBOT, and HPOT. All drugs induced rhythmic contraction. The average increase in contraction strength calculated from the AUC at 1 nM was 127 ± 5.4% for OT (n = 11), 99 ± 1.9% for carbetocin (n = 4), 142 ± 10.6% for FBOT (n = 13), and 166 ± 14.9% for HPOT (n = 8) (Fig 2A). One-way ANOVA, followed by multiple comparisons, showed significant differences ($F_{3,32}$ = 4.557, $P$ = 0.0091). Bonferroni's post hoc analysis revealed that contraction induced by HPOT was significantly larger than that induced by OT and carbetocin ($P < 0.05$).

The *ex vivo* dose-dependent activities of analogs for the human uterus are shown in Fig 2B. A curve was obtained by examining the effects of each agonist at concentrations ranging from 10 pM to 100 µM in single preparations in duplicate. The highest potency was obtained for HPOT (190 pM), followed by that for FBOT (556 pM), OT (5,340 pM), and carbetocin (12,090 pM) (Table 1).

From the same experiments shown in Fig 2B, the agonist efficacies compared to that for OT ($E_{max}$ = 100%) were found to be 86% for FBOT and 75% for HPOT, similar to that observed for carbetocin (81%) (Table 1). These data completely differ from those regarding OT receptor responses in human embryonic kidney (HEK) cells, in which FBOT elicited larger responses than OT and was hence annotated a superagonist [7].

Next, we examined sensitivity to the antagonist atosiban. As shown in Fig 3, compared to the control (without atosiban), atosiban (1 µM) reduced contraction, as was evident from the increase in AUC to 82 ± 6.3% (n = 13). Compared to that observed after atosiban pre-treatment, pre-treatment with 1 nM OT or OT analogs changed contraction strength in the

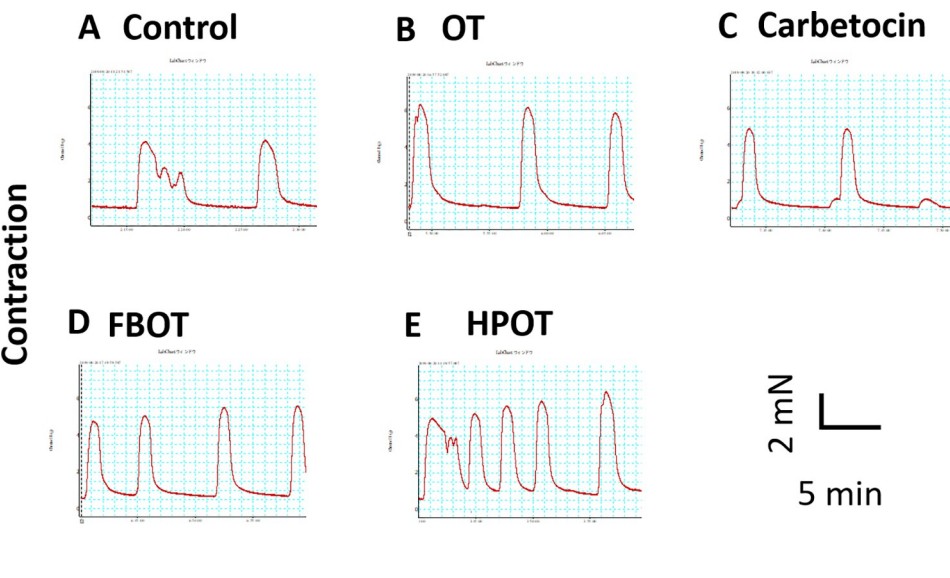

**Fig 1. Representative *ex vivo* contractility traces of myometrial strips isolated from human caesarean sections.** After establishing spontaneous rhythmic uterine contraction (A), effects of 1 nM oxytocin (OT, **B**), carbetocin (**C**), FBOT (**D**), and HPOT (**E**) on spontaneous rhythmic uterine contraction were examined. Contractions were recorded for 6 min. The time interval between drug application and washing was 30 min.

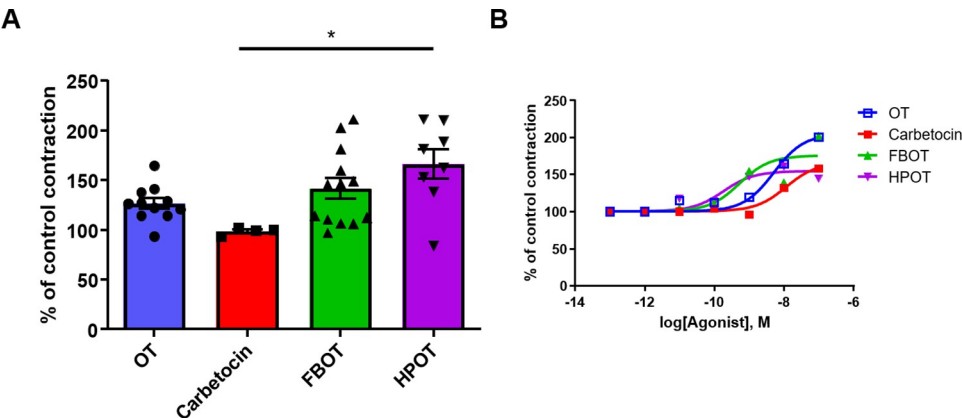

**Fig 2. Strength of human myometrial contractivity.** (**A**) Percentage of strength of contractivity in response to 1 nM oxytocin (OT), carbetocin, FBOT, and HPOT treatments, calculated from traces shown in Fig 1; 100% represents the area under curve (AUC) of control contractivity. *$P < 0.05$. (**B**) Dose response curves of OT, carbetocin, FBOT, and HPOT. Recordings of human myometrial contractivity (n = 4–13 for each point) were analyzed for AUC; 100% represents AUC of spontaneous contraction in the presence of 1 pM of each agonist. Note that symbols overlap at concentrations lower than 0.1 nM. Symbols of HPOT and FBOT overlap with those of OT at 10 and 100 nM, respectively.

following order: OT (119 ± 12%) > HPOT (114 ± 5.2%) > FBOT (96.3 ± 5.2%). These data suggested that HPOT and FBOT increased contraction after blocking of OT receptors by the antagonist, albeit in a different manner.

To determine the drug interactions among OT, FBOT, and HPOT, the response in the mixture was compared with the AUC obtained in a single application. Additivity was calculated using the Bliss independence model. The HPOT and OT mixture displayed interaction [confidence interval (CI) was 1.01], which was 176% of that of the control AUC and was similar to that observed with the single application of HPOT (166%) (**Table 2**). In contrast, the AUC of the OT-FBOT mixture was 140%, with CI of 1.45. These values indicated that FBOT showed infra-additivity and that the activation mode of FBOT was same as that of OT, while that of HPOT differed.

## Effects of OT analogs on contraction in uterine muscles of virgin mice

The agonistic effects of FBOT and HPOT were confirmed in the mouse uterus (**Fig 4**). The potency of contraction strength (**Fig 5**) was ranked based on $EC_{50}$ values as follows: FBOT (56 pM) > OT (239 pM) > carbetocin (2718 pM) > HPOT (3150 pM) (**Table 1**).

**Table 1. *Ex vivo* activities of oxytocin and analogs in human and mouse uteri.**

| Agonists | Human | | | Mouse | |
|---|---|---|---|---|---|
| | $EC_{50}$ (pM) | 95% Confidence interval | Emax (%) | $EC_{50}$ (pM) | 95% Confidence interval |
| Oxytocin | 5340 | 1809–15760 | 100 | 239 | 147–390 |
| Carbetocin | 12090 | 4816–30340 | 81 | 2718 | 1149–6429 |
| FBOT | 556 | 32–9749 | 86 | 56 | 22–138 |
| HPOT | 190 | 31–1163 | 75 | 3150 | 1102–9000 |

The effects on contraction were determined as the percentage of area under the curve (AUC) to the AUC of control contraction. Each experiment was performed in duplicate in two independent samples.

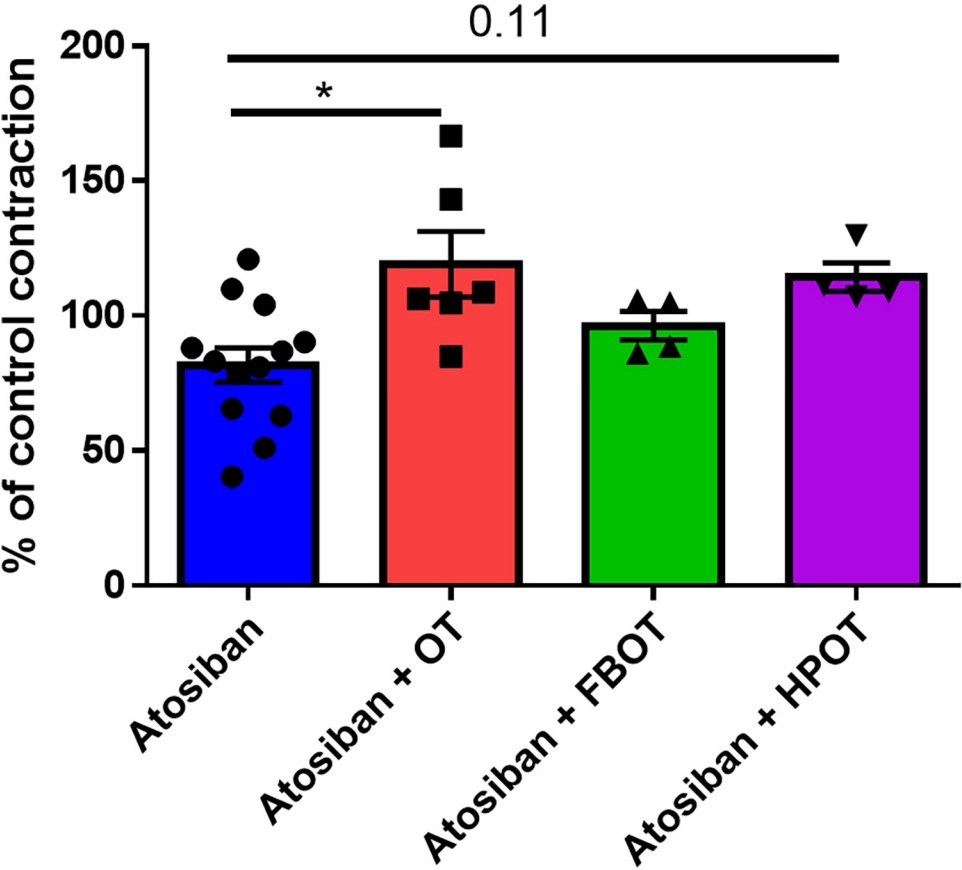

**Fig 3. Recovery of *ex vivo* human uterine contraction by OT and its analogs after atosiban-mediated inhibition.**
Contractions induced by 1 nM OT (OT; n = 6), FBOT (n = 4), and HPOT (n = 4) in the presence of 1 μM atosiban
(n = 13) in 2–6 independent samples with duplicate measurements. Percentage of contraction was calculated using
AUC. 100% represents spontaneous contraction without any treatment. Atosiban was applied 5 min before treatment
with agonists. $^*P < 0.05$.

## Discussion

The effects of two new OT analogs, FBOT and HPOT, on the AUC of myometrial contractivity
of human uterus samples and virgin mouse uterus were tested *ex vivo* and compared with
those of OT and carbetocin. Drug-drug interactions and the ability to overcome the effects of
an OT receptor antagonist were evaluated using the human uterus. FBOT and HPOT demonstrated higher sensitivity than OT or carbetocin, with higher activity at 1 nM concentration.
Interestingly, both compounds demonstrated partial agonistic effect at higher concentrations
of 100 nM. FBOT, which has been reported as compound 2 [7], demonstrated 86% of the agonist efficacy of OT in terms of AUC of human uterine contraction. The agonist efficacies of
carbetocin and HPOT, reported previously as compound 5 [7], was 75% and 81% of that of
OT, respectively. However, at 1 nM, both HPOT and FBOT showed the same contraction
strength as OT, demonstrating higher potency. In experiments on the human uterus, the
potency order observed was HPOT > FBOT > OT > carbetocin. These data suggested that
under *ex vivo* conditions, HPOT and FBOT behave as potent agonists for human uterine OT
receptors. These results are in agreement with the *in vitro* efficacy and potency of OT receptors
overexpressed by the human OT gene in HEK-293 cells [7,33].

**Table 2. Additivity using the Bliss independence model.**

| Agonist | Percentage of control AUC | EA | EB | EAB | CI |
|---|---|---|---|---|---|
| OT ($10^{-9}$ M) | 126 | 0.25 | | | |
| FBOT ($10^{-9}$ M) | 142 | | 0.4 | | |
| HPOT ($10^{-9}$ M) | 166 | | 0.63 | | |
| OT-FBOT ($2 \times 10^{-9}$ M) | 140 | | | 0.38 | 1.45 |
| OT-HPOT ($2 \times 10^{-9}$ M) | 176 | | | 0.72 | 1.01 |

The additivity of oxytocin (OT) and analogs was calculated according to the Bliss independence model using the formula CI = (EA + EB—EAxEB)/EAB. Response is considered in the range $0 < E < 1$, where 0 is percentage of control contraction AUC and 1 is percentage of Emax contraction AUC. Each experiment was performed in duplicate. EA–response for oxytocin application, EB–response for tested compound application and EAB–response for mix of oxytocin and compound.

Experiments on virgin mouse uterus showed that FBOT possesses strong agonistic effects, confirming its superagonist function in OT receptor-overexpressing HEK-293 cells [7]. In sharp contrast, HPOT showed the lowest $EC_{50}$ value among the four agonists tested in the same mouse tissue, while HPOT was the most potent agonist in human tissue. The reason for this discrepancy is unclear. This may be because of species-specific differences and differences between non-pregnant and pregnant tissues. Therefore, the effects of HPOT have to be assessed in the pregnant uterus or uterine muscles during parturition in mouse mothers. In the current research, we limited our study only by the inclusion of virgin mice in order to

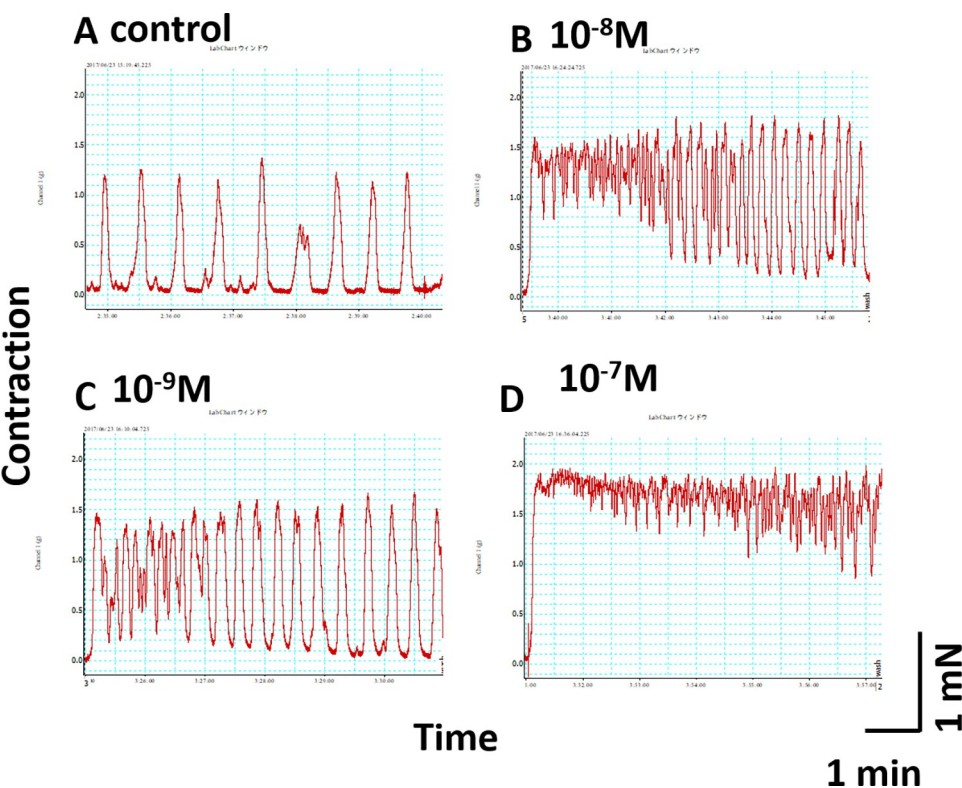

**Fig 4. Representative *ex vivo* contractility traces of myometrial strips isolated from virgin mice.** After spontaneous rhythmic uterine contraction, (**A**) effects of indicated concentrations of FBOT, ranging from 1 nM to 100 nM (**B-D**) were examined in the same muscle strip. Contractions were recorded for 6 min. The time interval between drug application and washing was 30 min.

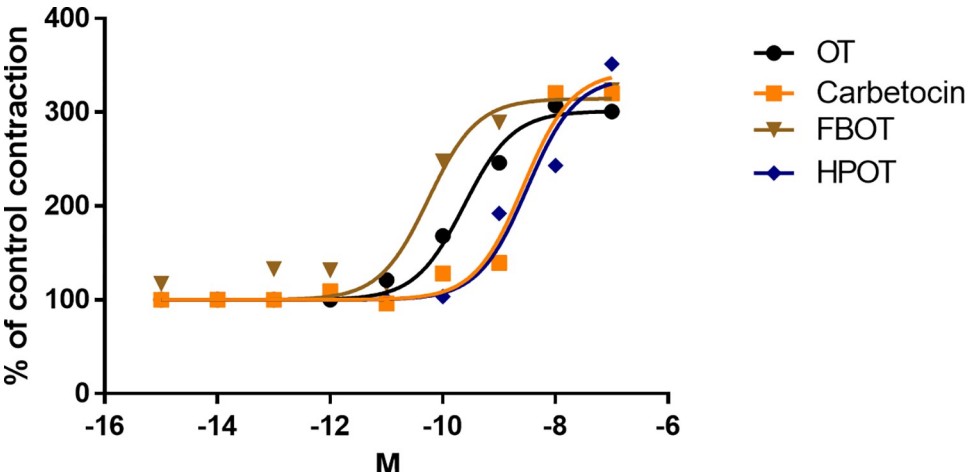

**Fig 5. Dose response curves showing the effects of OT, carbetocin, FBOT, and HPOT on strength of contraction of the mouse uterus.** Mouse myometrial contractility were recorded as shown in Fig 4 from seven virgin mice. Contraction strength was calculated from the area under the curve (AUC). n = 3–5 for each dot; 100% indicates AUC of spontaneous contraction.

obtain exactly the same tissue region for comparison, where similar muscle composition is expected.

Studies have shown that compared to that in non-pregnant myometrium, the mRNA level of OT receptors increases 100-fold at 32 weeks and >300-fold at the onset of parturition [34]. In addition, the level of OT receptor mRNA in the myometrium increases almost 50 times after 12 h of labor [35]. However, we assumed that the OT receptor mRNA level may not change considerably between 2 h and 6 h during in *ex vivo* experiments. However, as plasma OT levels before excision may also modulate myometrial sensitivity to OT and its analogs, OT receptor numbers may be modulated by desensitization (internalization).

We evaluated the ability of the OT receptor antagonist, atosiban, to overcome the agonistic effects of analogs. While pre-treatment with 1 μM atosiban decreased the level of spontaneous contractility, secondary treatment with 1 nM analogs partially recovered contractility in the following order: HPOT > OT > FBOT. Thus, the ability of the analogs to counter the effects of atosiban did not match the potency or affinity orders as atosiban is known to be a non-selective antagonist acting via OT receptors and arginine vasopressin V1A or V1B receptors [36], one of the main differences between OT and its analogs may be selectivity for the V1A receptors [7]. HPOT was demonstrated to be more selective in the activation of OT receptors, especially at lower concentrations, while FBOT activated V1A receptors at higher concentrations [7]. Many reports have shown abundant expression of V1A receptors in human and rodent myometrium and demonstrated that vasopressin plays a significant role in uterine contraction [37–39]. Therefore, the difference in the abilities to compete with the antagonist can be explained by the differences in the activation of V1A receptors by OT and its analogs. In contrast, some studies have suggested that the function of V1A receptors in uterine contraction is similar to that of OT receptors, suggesting that lower recovery by FBOT is possibly due to selective stimulation of OT receptors during uterine contraction [38].

Understanding whether FBOT, HPOT, and OT share the same types of receptors and the drug-drug interactions is critical. We tested these synergistic (additivity) or antagonistic effects by simultaneously applying OT and either of the two OT analogs. At a concentration of 1 nM which close to their Ki values [7], a mixture of 1 nM OT and 1 nM FBOT produced uterine

contraction higher than that induced by OT alone and lower than that induced by FBOT alone, suggesting that OT and FBOT antagonize each other and show infra-additivity [40,41]. Identical experiments demonstrated that contractivity induced by a combination of HPOT and OT was higher than that induced by application of either OT or HPOT, but CI close to 1 suggest absence of additivity.

From a clinical point of view, it is important to test different possible combinations of uterotonics including a combination of FBOT or HPOT and Carbetocin, or a combination of more than 2 uterotonics. This issue should be addressed in future studies.

Based on data from previous *in vitro* and the current *ex vivo* studies, we concluded that the new OT analogs are better than OT in inducing uterine contractions at lower concentrations. This feature can be beneficial in reducing the use of uterotonics for the control of blood loss during cesarean section. In addition, the analogs utilize different receptor kinetics as mentioned above [7] and their effects last longer than those of OT in recovery from social impairments. Obtained data suggests HPOT administration may immediately stop bleeding as an emergency uterotonic agent. In addition, only single administration of long-acting HPOT can facilitate contraction during delivery. More importantly, even if blood or tissue concentrations of FBOT increase due to accumulation after continuous administration, muscle contraction is limited because of the limiting ability of partial agonists to increase power of contraction. We can estimate a similar extent of the effects of FBOT. However, antagonistic drug-drug interactions of OT and FBOT should be considered when determining the combination and dose regimen. The high selectivity of both compounds for the human OT receptor [7] may be beneficial in reducing the possible side effects of the compounds; however, a comprehensive study of the possible side effects of these OT analogs is warranted.

Whether FBOT and HPOT exert vasoconstrictive effects via the activation of V1A receptors remains to be determined. Vasoconstriction is not suitable for delivery operations, as the fetus may be damaged by low $O_2$ supply. The extent to which HPOT and FBOT can elevate blood pressure warrants investigations. Until then, HPOT and FBOT should not be administered to patients with preeclampsia who may not tolerate abrupt increases in blood pressure. Whether long-lasting actions ($>15$ h) of HPOT and FBOT (HPOT lasts longer than FBOT) are more beneficial than OT (which lasts only for $1-2$ h) in clinical implications have to be considered. Simultaneous use of FBOT and OT displayed additivity, whereas identical effect was not observed with HPOT and OT. This also has to be considered when using HPOT and FBOT in combination treatment in clinical applications.

Despite the translational information obtained at the *ex vivo* level in this study, several questions regarding the *in vivo* usage of the analogs persist. First, we observed differences in the additivity of agonists in the uterine tissue, which interact not only with human OT receptors but also with several other receptors, including V1A receptors. Therefore, whether the analogs shared the same receptor with OT is still not clear. In future, we plan to use *in vitro* cellular assays for measuring drug-drug interactions using a wider dose range and more strict calculation methods such as Loewe additivity.

The receptor signaling used by the analogs requires investigation. Previously, we have only evaluated pathways related to Gq signaling by $Ca^{2+}$ mobilization for FBOT and HPOT [7,42]. In fact, it is known that higher concentrations of OT can activate Gi1-3, GoA, and GoB pathways [43] and OT can couple with ADP-ribosyl cyclase [3]. The contributions of FBOT and HPOT to such signaling pathways remain unclear.

The differences in the responses of uterine tissue pretreated with atosiban to the analogs cannot be explained by the analogs' affinity, potency, or efficacy. Previously, we have shown that the residence time of HPOT on OT receptors is higher than that of OT [7]. This may indicate that HPOT activity is based on blockade of antagonist inhibition.

Here, we demonstrated that both OT analogs acted on human uterine tissue with high potency and elicited strong contractive responses. However, the sample size of the human uterus from cesarean deliveries was too small to show statistically significant differences. In addition, pregnant terms and time of labor differed considerably within samples, which was difficult to control. An additional limitation was the gestation status. For human experiments, tissues were obtained from elective cesarean section at 30–40 weeks of pregnancy. Although the number of study cases collected from any university hospital is limited, our techniques for studying contraction in the human uterus were innovative, as we performed identical experiments on mouse uterus more than 50 times. Thus, technical variations were low. In addition, we attempted to use the muscle tissues of the fixed uterine region as much as possible (**S1 Fig**).

According to data obtained in human uterine samples, we designed to cut the longitudinal direction that is frequently used, because it is known that the maximum OT-induced contraction can be obtained in this direction in muscles (**S1 Fig**) [44]. However, the human uterine muscle shows regional variations in circular or longitudinal directions, being similar in the horns and corpus and lower in the cervix [45]. Compared to the potency of the analogs, this gradation in contractile responsiveness to OT will lead to inaccuracy of determination compound potency levels. However, in this study, we tried to reduce these variations by carefully excising the marginal portion of the incised uterus during the cesarean section.

The contraction amplitude was expected to correctly reflect the concentrations of OT analogs applied to the human uterine samples. To reduce variations, we selected the uterus at 36–38 weeks of pregnancy. However, we did not pay attention to the difference between primipara or multipara parities; 65% of the cases were primipara women, which may be another factor that influenced OT sensitivity. The complications and indications for cesarean deliveries differed considerably in each case, although this may not be surprising.

## Conclusion

In summary, our results show that FBOT and HPOT are stronger uterotonic and highly potent agents with better performance in the subnanomolar range than OT or carbetocin. This may be beneficial for controlling postpartum uterine tone, which remains an important subject in obstetric and anesthesiology. The lack of proper guidelines regarding OT usage during parturition indicates that the possible side effects originating from the use of large amounts of OT still remain an important issue. The current OT analogs may reduce side effects, as HPOT and FBOT act at lower concentration ranges than OT and carbetocin. In future, the effects of these analogs have to be tested *in vivo* in clinical studies on labor induction, constriction of uterus showing atony, cessation of bleeding, vasoconstriction, and high blood pressure.

## Supporting information

**S1 Checklist. The ARRIVE guidelines 2.0: Author checklist.**
(PDF)

**S1 Fig. Scheme of uterine incision.**
(TIF)

**S1 Table. Characteristic of patients.**
(TIF)

## Author Contributions

**Conceptualization:** Stanislav M. Cherepanov, Takashi Hosono, Masanori Ono, Hiroshi Fujiwara, Satoshi Shuto, Haruhiro Higashida.

**Data curation:** Stanislav M. Cherepanov, Takashi Iizuka, Takashi Hosono, Satoshi Shuto, Haruhiro Higashida.

**Formal analysis:** Stanislav M. Cherepanov.

**Funding acquisition:** Haruhiro Higashida.

**Investigation:** Stanislav M. Cherepanov, Teruko Yuhi, Takashi Iizuka, Satoshi Shuto.

**Methodology:** Stanislav M. Cherepanov, Teruko Yuhi, Takashi Iizuka, Masanori Ono, Hiroshi Fujiwara, Satoshi Shuto, Haruhiro Higashida.

**Project administration:** Stanislav M. Cherepanov.

**Resources:** Satoshi Shuto, Haruhiro Higashida.

**Supervision:** Masanori Ono, Shigeru Yokoyama, Haruhiro Higashida.

**Validation:** Masanori Ono, Hiroshi Fujiwara, Shigeru Yokoyama, Haruhiro Higashida.

**Visualization:** Stanislav M. Cherepanov.

**Writing – original draft:** Stanislav M. Cherepanov.

**Writing – review & editing:** Stanislav M. Cherepanov, Haruhiro Higashida.

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
