## [Decision Letter · Decision Letter 0]

8 Nov 2022

PONE-D-22-23679Two oxytocin analogs, N-(p-fluorobenzyl) glycine and N-(3-hydroxypropyl) glycine, induce uterine contractions ex vivo in ways that differ from that of oxytocinPLOS ONE

Dear Dr. Cherepanov,

Thank you for submitting your manuscript to PLOS ONE. After careful consideration, we feel that it has merit but does not fully meet PLOS ONE’s publication criteria as it currently stands. Therefore, we invite you to submit a revised version of the manuscript that addresses the points raised during the review process.

We look forward to receiving your revised manuscript.

Kind regards,

David Desseauve, MD, MPH, PhD

Academic Editor

PLOS ONE

Journal Requirements:

2. As part of your revision, please complete and submit a copy of the Full ARRIVE 2.0 Guidelines checklist, a document that aims to improve experimental reporting and reproducibility of animal studies for purposes of post-publication data analysis and reproducibility: https://arriveguidelines.org/sites/arrive/files/documents/Author%20Checklist%20-%20Full.pdf  (PDF). Please include your completed checklist as a Supporting Information file. Note that if your paper is accepted for publication, this checklist will be published as part of your article

4. We note that you have stated that you will provide repository information for your data at acceptance. Should your manuscript be accepted for publication, we will hold it until you provide the relevant accession numbers or DOIs necessary to access your data. If you wish to make changes to your Data Availability statement, please describe these changes in your cover letter and we will update your Data Availability statement to reflect the information you provide

Reviewers' comments:

Reviewer's Responses to Questions

**Comments to the Author**

1. Is the manuscript technically sound, and do the data support the conclusions?

Reviewer #1: Yes

Reviewer #2: Yes

2. Has the statistical analysis been performed appropriately and rigorously? 

Reviewer #1: Yes

Reviewer #2: Yes

3. Have the authors made all data underlying the findings in their manuscript fully available?

Reviewer #1: Yes

Reviewer #2: Yes

4. Is the manuscript presented in an intelligible fashion and written in standard English?

Reviewer #1: Yes

Reviewer #2: Yes

5. Review Comments to the Author

Reviewer #1: Thank you for this great work.

It seems to me that it should be made clear why virgin mouse uteri and muscle fibers from pregnant women were chosen. Why did you not make a comparison between uteri of virgin mice and uteri of mice that were already pregnant? Why didn't they collect uterine muscle fibers from women who had never been pregnant?

Furthermore, it is not specified on the women from whom uterine muscle fibers were taken whether it was a first or second caesarean section. Indeed, during a second caesarean section, the risk is to take fibrous tissue which would create a bias during the measured contractions.

Reviewer #2: Thank you for allowing me to review “Two oxytocin analogs, N-(p-fluorobenzyl) glycine and N-(3-hydroxypropyl) glycine, induce uterine contractions ex vivo in ways that differ from that of oxytocin »

PPH is still an issue wordwide, and research is essential to find new safe and cost effective utero tonic drug.

Here are some comment:

Abstract :

Please reconsider the last sentence “they can be used as potent inducers of labor”

Concentrations of uterotonic are significantly lower in the induction of labor and this study was conduct to reduce the PPH. Thus you cannot conclude your abstract with this sentence.

Introduction

« data obtained from laboring and non-laboring parturients have shown that carbetocin a long acting… »

In deed, the carbetocin dont need an additionnal uterotonic but the PPH effect is not that remarkable compared to other uterotonic especially regarding the cost-effectiveness

Method

Line 136 “Women 136 with multiple pregnancies or medical conditions, such as diabetes, preeclampsia, or 137 obstetric cholestasis, were not included in this study »

Did you consider the BMI of patient?

Line 147 “Myometrial biopsies were excised 147 within 3 min of OT administration. »

Was it compromising that the human myometrial biopsie was realized 3 min after 5UI of OT?

Result

Line 211 “Effects of oxytocin analogs on contraction of human uterine muscles »

How do yo explain that carbetocin induced less contraction than OT?

Discussion

Line 382 “Simultaneous use of FBOT 385 and OT displayed additivity, whereas identical effect was not observed with HPOT and 20 386 OT. »

The management of PPH often involves the use of a combination of uterotonic drog. Did you evaluate the combination: FBOT+ OT and HPOT+ OT and carbotacin

Line 423 “However, in this 424 study, we tried to reduce these variations by carefully excising the marginal portion of 425 the incised uterus during the cesarean section. »

An other limit should be notice. The myometrial biopsies were excided in the transverse uterin incision, witch correspond to the uterine isthmus. This region is thinner and contains then less uterine muscle. Those results could be underestimated

6. PLOS authors have the option to publish the peer review history of their article (what does this mean?). If published, this will include your full peer review and any attached files.

Reviewer #1: No

Reviewer #2: No

---

## [Author Response · Author response to Decision Letter 0]

19 Dec 2022

Comment: Reviewer #1: Thank you for this great work. It seems to me that it should be made clear why virgin mouse uteri and muscle fibers from pregnant women were chosen. Why did you not make a comparison between uteri of virgin mice and uteri of mice that were already pregnant? 

Response: Thank you very much for reasonable comments and questions. 

You right about necessity of comparison of virgin vs pregnant uterine tissue, we address our discrepancy for HPOT, and use of pregnant uterine tissue can be useful in this context. However, it is also important to obtain the same tissue region, because the uterus is composed of heterogeneous muscles. Usage of the virgin mouse uterus is to obtain from the same region, in which similar muscle composition is expected and it is more complicated for pregnant mice. So, for this step we limited only for virgin mice for better precision. To address this issue, we modified paragraph in discussion as

“The reason for this discrepancy is unclear. This may be because of species-specific differences and differences between non-pregnant and pregnant tissues. Therefore, the effects of HPOT have to be assessed in the pregnant uterus or uterine muscles during parturition in mouse mothers. In the current research, we limited our study only by the inclusion of virgin mice in order to obtain exactly the same tissue region for comparison, where similar muscle composition is expected.” 

Comment: Why didn't they collect uterine muscle fibers from women who had never been pregnant?

Response: We considered human pregnant uteri to be the most appropriate for evaluating the effects of OT, carbetocin, FBOT, and HPOT treatments. For this reason, we received approval from the Ethics Committee of Kanazawa University, and obtained consent from pregnant women undergoing caesarean section to provide specimens. Because in that case caesarean section was appropriate procedure for patients, it’s possible to obtain approval. Collection of uterine tissue from healthy non-pregnant woman will be ethically unjustified. 

Comment: Furthermore, it is not specified on the women from whom uterine muscle fibers were taken whether it was a first or second caesarean section.

Response: Thanks for notice. We add next line to the methods description: 

“There were 3 secondary and 20 primary cesarean sections out of 23.”

Comment: Indeed, during a second caesarean section, the risk is to take fibrous tissue which would create a bias during the measured contractions.

Response: For the second caesarean section, we performed a caesarean section with an incision cranial to the uterine incision scar of the previous caesarean section. For this reason, we were able to carry out the research with very little contamination of fibrous tissue. 

We add next sentence to method section: 

“In the case of the secondary caesarean section, we performed a caesarean section with an incision cranial to the uterine incision scar of the previous caesarean section to minimize the presence of fibrous tissue”

Comment: Reviewer #2: Thank you for allowing me to review “Two oxytocin analogs, N-(p-fluorobenzyl) glycine and N-(3-hydroxypropyl) glycine, induce uterine contractions ex vivo in ways that differ from that of oxytocin » PPH is still an issue wordwide, and research is essential to find new safe and cost-effective utero tonic drug.

Response: Thank you very much for your review, interest to our research and good points.

Comment: Abstract :

Please reconsider the last sentence “they can be used as potent inducers of labor”

Concentrations of uterotonic are significantly lower in the induction of labor and this study was conduct to reduce the PPH. Thus you cannot conclude your abstract with this sentence.

Response: Thank you for your valuable comment. As you pointed out, we have deleted this sentence.

Comment: Introduction

« data obtained from laboring and non-laboring parturients have shown that carbetocin a long acting… »

Indeed, the carbetocin dont need an additionnal uterotonic but the PPH effect is not that remarkable compared to other uterotonic especially regarding the cost-effectiveness

Response: Thank you very much for your valuable comment. That’s another point, why we still should look for new cost-effective uterotonics.

Comment: Method

Line 136 “Women 136 with multiple pregnancies or medical conditions, such as diabetes, preeclampsia, or 137 obstetric cholestasis, were not included in this study »

Did you consider the BMI of patient?

Response: No, we did not take in account BMI in current study

Comment: Line 147 “Myometrial biopsies were excised 147 within 3 min of OT administration. »

Was it compromising that the human myometrial biopsie was realized 3 min after 5UI of OT?

Response: Thank you for good question. The clinical benefits of adding oxytocin to caesarean sections are clear and ethically unavoidable. After removal of the specimen, it is immediately immersed in PBS in the operating room, and we believe that the effects of remaining oxytocin are minimal.

Comment: Result

Line 211 “Effects of oxytocin analogs on contraction of human uterine muscles »

How do yo explain that carbetocin induced less contraction than OT?

Response: Considering pharmacodynamics, Carbetocin is partial agonist of Oxytocin receptors with efficacy (Emax) lower than oxytocin (full agonist), this is was confirmed in previous studies on cell lines in vitro and confirmed here ex vivo. So, carbetocin can not activate as much receptors as oxytocin and unable produce same amount of contraction regardless of dosage. However, carbetocin have long-lasting activity, which beneficial in clinical context.

Comment: Discussion

Line 382 “Simultaneous use of FBOT 385 and OT displayed additivity, whereas identical effect was not observed with HPOT and 20 386 OT. »

The management of PPH often involves the use of a combination of uterotonic drog. Did you evaluate the combination: FBOT+ OT and HPOT+ OT and carbotacin

Response: Thank you for good point. We tested ex vivo interaction of FBOT+OT and HPOT+OT (Table 2), but did not test a lot of possible combinations such are combinations of Carbetocin+FBOT, Carbetocin+HPOT or FBOT+ OT and HPOT+ OT and carbotocin. While its is clinically important, there is limitation in our study. We address it in discussion and add next line:

“From a clinical point of view, it is important to test different possible combinations of uterotonics including a combination of FBOT or HPOT and Carbetocin, or a combination of more than 2 uterotonics. This issue should be addressed in future studies. “

Comment: Line 423 “However, in this 424 study, we tried to reduce these variations by carefully excising the marginal portion of 425 the incised uterus during the cesarean section. »

An other limit should be notice. The myometrial biopsies were excided in the transverse uterin incision, witch correspond to the uterine isthmus. This region is thinner and contains then less uterine muscle. Those results could be underestimated

Response: In our case myometrial biopsies were obtained from the upper margin of the incision made at the lower segment of the uterus.

---

## [Decision Letter · Decision Letter 1]

23 Jan 2023

Two oxytocin analogs, N-(p-fluorobenzyl) glycine and N-(3-hydroxypropyl) glycine, induce uterine contractions ex vivo in ways that differ from that of oxytocin

PONE-D-22-23679R1

Dear Dr. Cherepanov,

We’re pleased to inform you that your manuscript has been judged scientifically suitable for publication and will be formally accepted for publication once it meets all outstanding technical requirements.

Kind regards,

David Desseauve, MD, MPH, PhD

Academic Editor

PLOS ONE

Reviewers' comments:

Reviewer's Responses to Questions

**Comments to the Author**

1. If the authors have adequately addressed your comments raised in a previous round of review and you feel that this manuscript is now acceptable for publication, you may indicate that here to bypass the “Comments to the Author” section, enter your conflict of interest statement in the “Confidential to Editor” section, and submit your "Accept" recommendation.

Reviewer #1: All comments have been addressed

Reviewer #2: All comments have been addressed

2. Is the manuscript technically sound, and do the data support the conclusions?

Reviewer #1: Yes

Reviewer #2: Yes

3. Has the statistical analysis been performed appropriately and rigorously? 

Reviewer #1: Yes

Reviewer #2: Yes

4. Have the authors made all data underlying the findings in their manuscript fully available?

Reviewer #1: Yes

Reviewer #2: Yes

5. Is the manuscript presented in an intelligible fashion and written in standard English?

Reviewer #1: Yes

Reviewer #2: Yes

6. Review Comments to the Author

Reviewer #1: Thank you for your answers.I don't have any other question. I think this paper is suitable for publication.

Reviewer #2: (No Response)

7. PLOS authors have the option to publish the peer review history of their article (what does this mean?). If published, this will include your full peer review and any attached files.

Reviewer #1: No

Reviewer #2: No

---

## [Editor Report · Acceptance letter]

30 Jan 2023

PONE-D-22-23679R1 

Two oxytocin analogs, N-(p-fluorobenzyl) glycine and N-(3-hydroxypropyl) glycine, induce uterine contractions ex vivo in ways that differ from that of oxytocin. 

Dear Dr. Cherepanov:

I'm pleased to inform you that your manuscript has been deemed suitable for publication in PLOS ONE. Congratulations! Your manuscript is now with our production department. 

Kind regards, 

on behalf of

Dr. David Desseauve 

Academic Editor

PLOS ONE